# Performance and Mechanism of Hydrothermally Synthesized MoS₂ on Copper Dissolution

**Hao Lu** [1], **Fang Cao** [1], **Xiaoyu Huang** [2] **and Honggang Yang** [1],*

1  School of Resources and Environmental Engineering, Wuhan University of Technology, Wuhan 430070, China
2  School of Management, Wuhan University of Technology, Wuhan 430070, China
*  Correspondence: yhg830218@whut.edu.cn

**Abstract:** The recovery of copper from circuit boards is currently a hot topic. However, recycling copper from circuit boards economically and environmentally is still a considerable challenge. In this study, a simple hydrothermal method was used to synthesize MoS₂ with nano-flower-like morphology using sodium molybdate dihydrate and thiourea as molybdenum and sulfur sources. The metal copper in the chip was successfully dissolved under the action of free radicals produced by ultrasound. The results show that under the catalytic action of hydrothermal synthesis MoS₂, the concentration of $Cu^{2+}$ dissolved by ultrasonic treatment for 10 h is 39.46 mg/L. In contrast, the concentration of $Cu^{2+}$ dissolved by commercial MoS₂ is only 2.20 mg/L under the same condition. The MoS₂ is polarized by external mechanical forces and reacts with water to produce $H^+$ and free electrons $e^-$, which can combine with $O_2$ and $OH^-$ to produce $\cdot OH$ and $\cdot O_2^-$ free radicals. Elemental Cu is converted to $Cu^{2+}$ by the attack of these two free radicals.

**Keywords:** molybdenum disulfide; piezoelectric catalysis; copper dissolution





## 1. Introduction

Reasonable recovery of electronic waste is a hot research topic in metal recycling [1,2]. At present, there are generally four methods for metal recovery treatment: mechanical treatment, wet treatment, microbial treatment, and fire treatment [3–5]. Mechanical processing requires the shredding of circuit boards, resulting in shredding costs and dust contamination [6]. Wet treatment can reduce crushing costs and avoid secondary dust pollution, but the operation is complicated and inefficient [7,8]. The microbial treatment cycle is long and easily affected by the environment [9]. The production temperature of fire treatment is as high as 1000 °C, the production investment is significant, and the energy consumption is excessive. It can be seen that current recycling methods have shortcomings such as low efficiency, narrow scope of application, and significant investment in production. Therefore, finding more green, efficient, and sustainable methods is the focus of current research.

Recently, Chen et al. used TiO₂ photocatalytic materials [10] under the action of photocatalysis. By means of this method, seven precious metals such as gold, silver, platinum, and rhodium in the circuit board; the ternary catalytic converter; and ore were selectively recovered. The noble metal with high purity (more than 98%) was recovered by a simple reduction reaction with acetonitrile and methylene chloride as extraction solution without using strong acid and base. The study proved that the methyl radical and superoxide radical were the active oxidizing species in the whole reaction system, and the recovery effect of more than 1 kilogram was also achieved. Muscetta et al. reported the use of ethanol as an anodic sacrifice. In their study, UV-visible light irradiation, solution pH = 6 conditions [11], the catalytic effects of different catalysts (TiO₂, WO₃ and ZnO), different catalyst loads, and different light irradiation conditions were studied by the method of ZnO catalytic deposition of palladium in a leaching solution. The results showed that ZnO had the best catalytic effect. When the catalyst load is less than 500 ppm, the relationship

between the catalyst load and the catalytic effect is linear. Compared with dark and single visible light conditions, UV-visible light irradiation has the best catalytic effect.

At present, piezoelectric materials have been widely studied in the fields of pollutant degradation, water cracking to produce hydrogen, heavy metal treatment, and tooth whitening [12–15]. For example, Qing et al. synthesized ultrathin zinc oxide nanosheets by a sodium dodecyl benzene sulfonate intercalation method [16]; ultra-thin $ZnO/Al_2O_3$ nanosheets were formed by calcination at high temperature, and methyl orange (MO), tetracycline (TC), and rhodamine B (RhB) were degraded under the action of ultrasound, respectively. The results show that the deformation of $ZnO/Al_2O_3$ nanosheets and the free charge related to oxygen-rich vacancies on the material surface contribute to the piezoelectric catalytic activity of the material. Pan et al. prepared $BaTiO_3$/graphene to effectively treat wastewater from CU-EDTA. The wastewater treated by this method will not generate secondary pollutants, which provides a new measure for the treatment of wastewater containing heavy-metal complexes. Wang et al. synthesized a piezoelectric nanoparticle $BaTiO_3$ with an average particle size of 130 nm to replace conventional toothpaste [17]; combined with mechanical vibration during brushing, the teeth cleaning process under piezoelectric catalysis was simulated. The results showed that this method significantly affected the removal of indigo carmine and rhodamine B stains within 3 h, and the teeth could be completely whitened within 10 h.

In this study, the copper in the circuit board was dissolved by piezoelectric catalytic technology without adding any strong acid, strong base, or toxic reagent. The dissolution rate is admittedly less optimistic than the traditional method of injecting stronger acids and bases and chemicals, but from the perspective of economy and environmental protection it is a viable solution that saves costs and does not pollute the environment. Until now, there have been few reports on the combination of piezoelectric catalysis and metal dissolution. In this paper, the hydrothermal synthesis of $MoS_2$ piezoelectric catalyst and ultrasonic-assisted free radical copper dissolution solution is expected to provide a new economic and sustainable development idea for the dissolution and recovery of scrap metal.

## 2. Results and Discussion

### 2.1. Analysis of Catalyst Characterization Results

#### 2.1.1. XRD and Raman Analysis of Catalysts

Figure 1a shows the X-ray diffraction patterns of commercial $MoS_2$ and synthetic $MoS_2$ samples. It can be seen from Figure 1a that the synthesized $MoS_2$ nanoflowers are single-phase $MoS_2$ and that the commercial $MoS_2$ has a sharp peak at $2\theta = 14°$, while the synthesized $MoS_2$ has a weaker peak intensity but a larger peak width here. Thus, it can be seen that the synthesized $MoS_2$ has a large number of monolayers and few layers [18]. At the same time, no other miscellaneous peaks appear in the XRD pattern of the synthesized $MoS_2$ catalyst, indicating that the $MoS_2$ has been successfully prepared and does not contain any other impurities. Figure 1b shows the corresponding Raman spectra. It can be seen from the spectra that there are two different peaks at about 380 and 405 cm$^{-1}$. The wavelength of the Raman spectra is broad and the intensity is low, which corresponds to the in-plane $E^1_{2g}$ and out-of-plane $A_{1g}$ vibration modes of $MoS_2$.

#### 2.1.2. SEM and TEM Analysis of Catalysts

Figure 2 shows the SEM images of commercial $MoS_2$ and synthetic $MoS_2$. It can be seen from the figure that commercial $MoS_2$ (Figure 2a,b) exhibits an irregular bulk structure with thick edges. Synthetic $MoS_2$ (Figure 2c,d) is mainly large and petal-like and has thinner edges than commercial $MoS_2$. In general, the thickness of the catalytic material and the amount of the edge have a direct impact on the piezoelectric catalytic effect in the reaction system, and the catalytic material with a multi-activity ultra-thin edge has better piezoelectric catalytic activity. As can be seen from Figure 2e, the spacing of the lattice fringes at the (002) crystal plane of the catalyst is very close to that reported in the literature, which is 0.64 nm [19]. Moreover, the catalyst has a large number of few-layers

and single-layers. As can be seen from the inset in Figure 2f, the nanoleaf-like catalyst is composed of a large number of tiny petals, which increases the catalytic active sites in the reaction system.

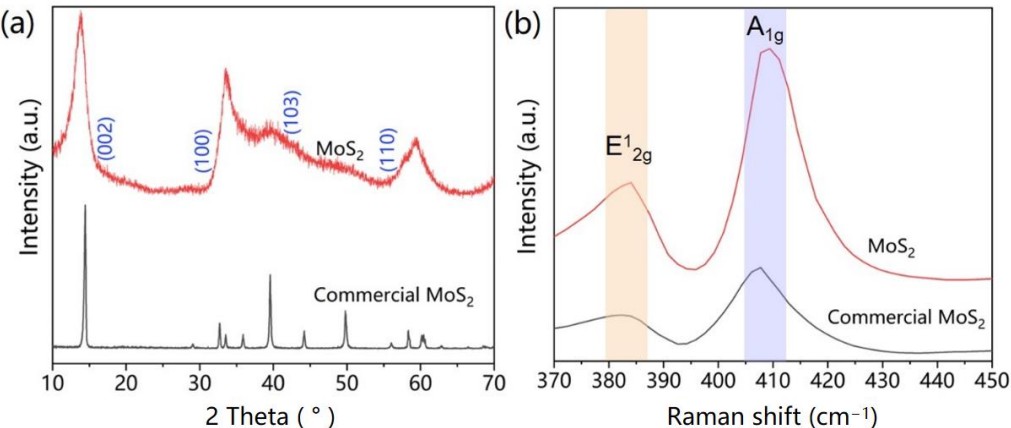

**Figure 1.** (**a**) XRD patterns of Commercial $MoS_2$ and $MoS_2$. (**b**) Raman spectra of Commercial $MoS_2$ and $MoS_2$.c

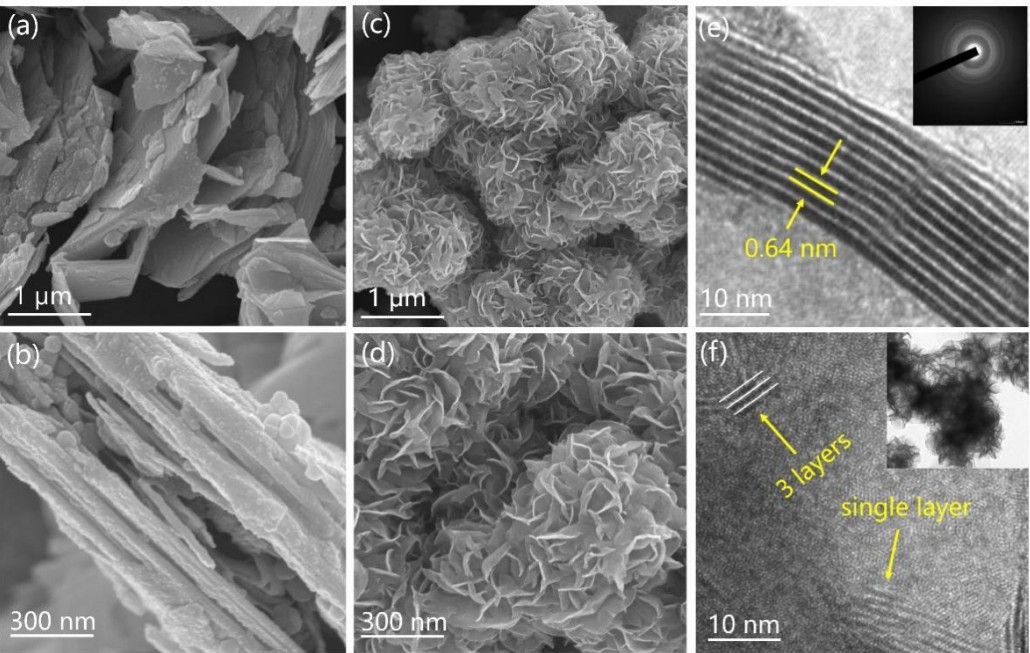

**Figure 2.** (**a**,**b**) SEM images of Commercial $MoS_2$. (**c**,**d**) SEM images of $MoS_2$. (**e**,**f**) TEM images of $MoS_2$.

### 2.1.3. PFM Analysis of Catalysts

In addition, piezoelectric microscopy was used to test the piezoelectric properties of the catalyst. Figure 3a shows the morphology of the sample, and the catalyst presents a nanoflower thickness less than 20 nm. At the same time, as shown in Figure 3b,c, the typical butterfly curve and piezoelectric reverse hysteresis loop of the catalyst are obtained under the DC-biased electric field of ±10 V. The test results show that the catalyst displacement is large and that the phase change is 180°, which further verifies that the catalyst has good piezoelectric catalytic performance.

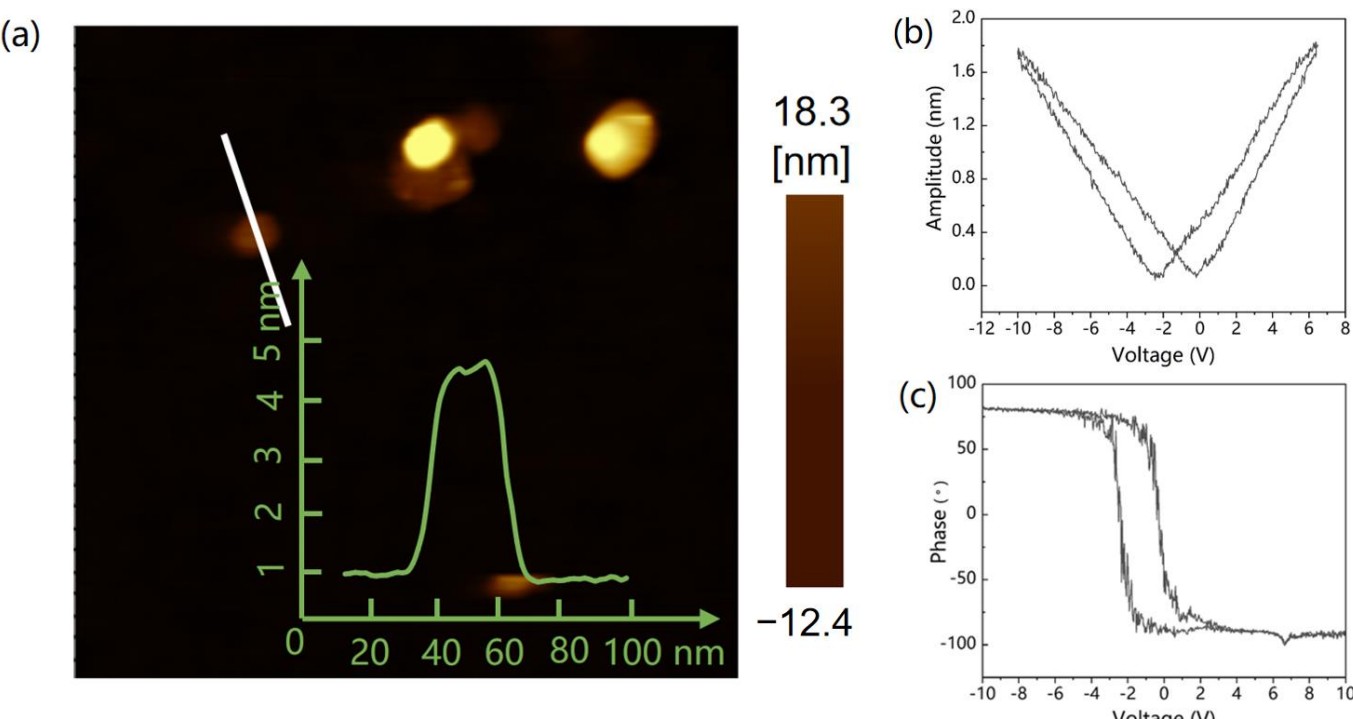

**Figure 3.** (**a**) AFM spectrum of MoS$_2$. (**b**) amplitude butterfly loops of MoS$_2$. (**c**) piezoelectric reverse hysteresis loop of MoS$_2$.

### 2.1.4. BET Analysis of Catalyst

Figure 4 shows the BET test of the commercial MoS$_2$ and synthetic MoS$_2$. It can be seen from Figure 4a that these two catalysts are in line with type III N$_2$ adsorption isotherms, that the adsorption is relatively weak, and that the rapid rise of the catalyst to near P/P$_0$ = 1 proves the existence of macropores [20]. The desorption curve is consistent with the characteristics of multilayer desorption and shows a small H$_3$ hysteresis loop. In addition, Figure 4b shows that the specific surface area of the commercial MoS$_2$ and synthetic MoS$_2$ is 3.20 and 27.48 m$^2$/g and that the pore volume is 0.016 and 0.155 m$^3$/g, respectively; this indicates that the synthetic MoS$_2$ had larger specific surface area and pore volume and thus better piezoelectric catalytic performance.

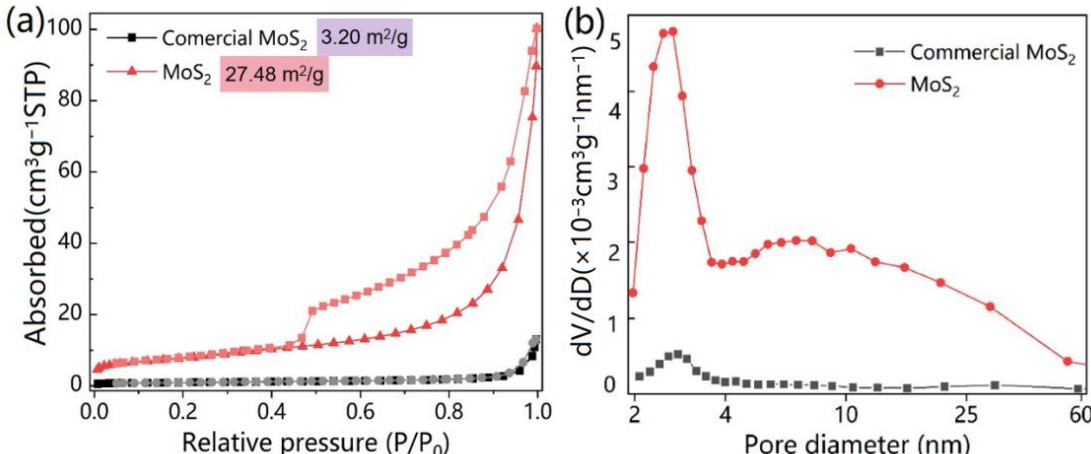

**Figure 4.** (**a**) N$_2$ adsorption–desorption isotherm. (**b**) pore size distribution of MoS$_2$ and commercial MoS$_2$.

*2.2. MoS$_2$ Catalyst for the Extraction of Copper in Cu Powder*

2.2.1. The Effect of Catalyst on Copper Dissolution

In order to investigate the effect of free radicals generated by MoS$_2$ catalyst under sonication on copper dissolution, the study of piezoelectric-catalyzed dissolution of copper was carried out at room temperature. By controlling the circulating water, the temperature of the water in the ultrasonic cleaner was kept in 24–30 °C. The results of the experiment are shown in Figure 5. Figure 5a demonstrates the effect of copper dissolution under different conditions. The concentration of dissolved copper was only 0.50 mg/L after 10 h of stirring under the synthesized MoS$_2$ catalyst and a magnetic stirrer with a speed of 400 r/min only; this was probably due to the weak mechanical vibration under stirring. Under the system with no catalyst, the solubility of the ultrasonic effect 10 h was only 1.37 mg/L; the dissolution of the copper in this case was due to the ultrasonic effect of producing a small amount of free radicals, which were weak in intensity but stronger than the mixing conditions. Under the joint action of commercial MoS$_2$ and ultrasonic for 10 h, the Cu$^{2+}$-dissolved concentration was 2.20 mg/L. Commercial MoS$_2$ has relatively poor piezoelectric catalytic performance due to its thicker layer and smaller specific surface area and pore volume; consequently, only a small amount of Cu$^{2+}$ dissolves. However, the synthetic MoS$_2$ can dissolve 39.46 mg/L Cu$^{2+}$ under the action of ultrasound for 10 hours+. Under this condition, the synthesized MoS$_2$ is prone to polarization under the action of ultrasound, producing e$^-$ and H$^+$, and producing ·O$_2$$^-$ and ·OH under the action of O$_2$ and OH$^-$, then oxidizing Cu to Cu$^{2+}$. Meanwhile, the dissolution process of copper under these conditions was investigated (Figure 6b). The results showed that the dissolved amount of Cu$^{2+}$ was 10.03, 24.81 and 39.82 mg/L after 1 h, 5 h and 10 h of sonication, respectively, which means that the dissolved amount of Cu$^{2+}$ in the system gradually increased with the increase in reaction time within 10 h of sonication.

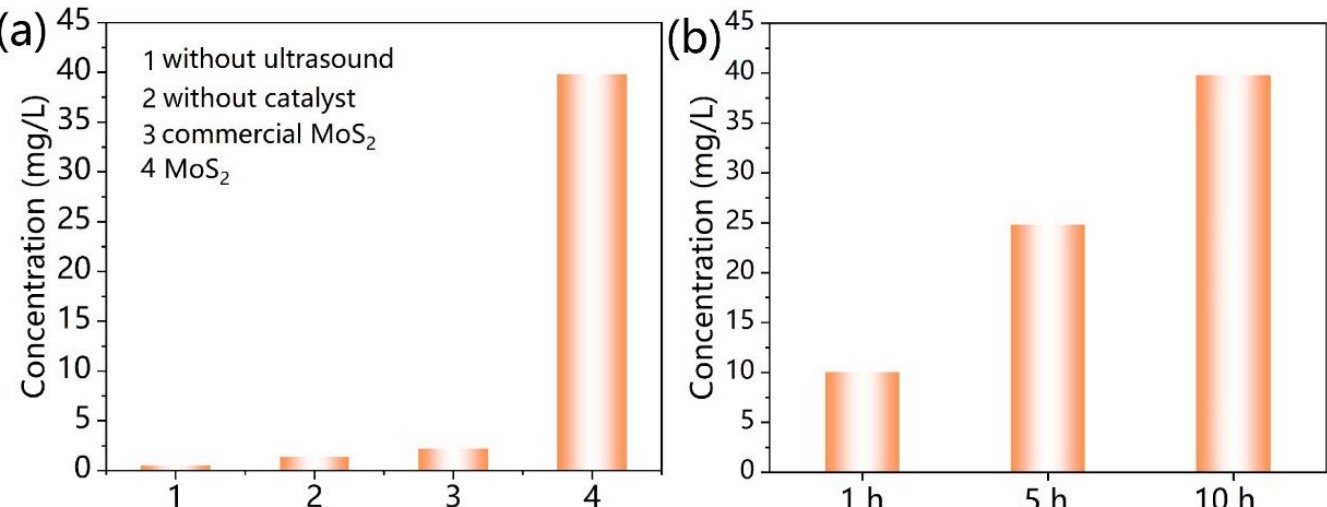

**Figure 5.** (**a**) Comparison of the effect of dissolving Cu$^{2+}$ under different conditions. (**b**) The change in Cu$^{2+}$ concentration with time.

2.2.2. Influence of pH and Different Gas Conditions on Copper Dissolution

pH is the critical factor affecting copper dissolution. As shown in Figure 6a (the experiment under this system is consistent with the above experimental conditions, and the ultrasonic action duration is 10 h), the more acidic the system environment is, the more prominent the dissolution effect of Cu$^{2+}$ is; this is consistent with the mechanism of metal ion leaching from acidic solution. In addition, this experiment studied the influence of different gas conditions in the reaction system on the dissolution amount of the copper. As shown in Figure 6b, the concentration of O$_2$ in the system promoted the dissolution of the copper, and the higher the flow rate into O$_2$, the higher the concentration of Cu$^{2+}$ dissolved,

because $O_2$ could be formed by combining with $e^-$ and $O_2^-$ radical, leading to an increase in the concentration of $\cdot O_2^-$ radical in the system. However, the introduction of $N_2$ into the system inhibits the dissolution of copper, and the larger the flow of $N_2$, the more obvious the inhibition effect. This may be because the intervention of $N_2$ reduces the collision between $e^-$ and $O_2$, and since the reaction solution is only 100 mL, the introduction of $N_2$ makes less $O_2$ soluble in the solution; consequently, the concentration of $\cdot O_2^-$ radical generated is lower.

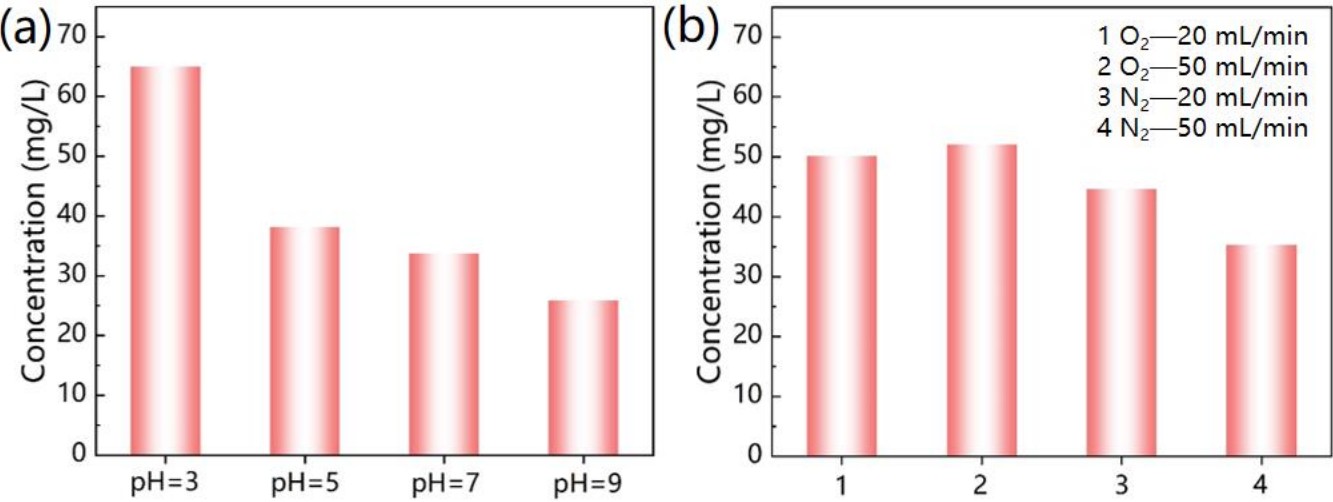

**Figure 6.** (**a**) effect of pH on copper dissolution. (**b**) effects of different gases on copper dissolution.

### 2.2.3. Diagram of the Dissolution Process of Copper

Figure 7 shows the process of copper dissolution. The upper suspension was collected after the reaction solution was left to precipitate for 12 h. The solution was separated by centrifugation in a centrifuge at 8000 r/min, and the $MoS_2$ catalyst was gathered in the centrifuge tube due to centripetal force. The clear solution obtained was placed in a beaker and filtered with a 0.22 μm filter head, i.e., a pure solution of $Cu^{2+}$ was obtained. Due to the low concentration of $Cu^{2+}$, the solution showed no color. After collecting the solution several times, it was be dried at 60 °C to obtain new copper.

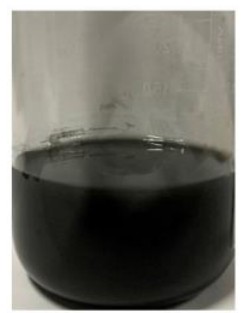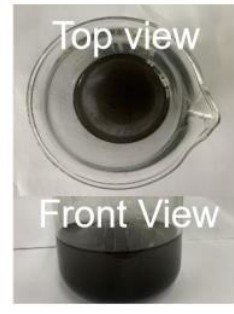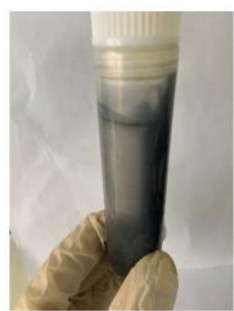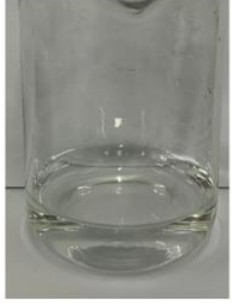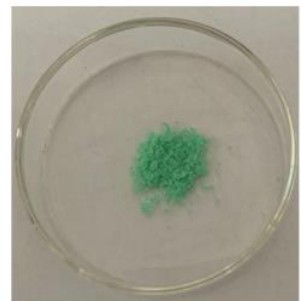

**Figure 7.** Diagram of the dissolution process of copper.

### 2.3. Dissolution Mechanism of $Cu^{2+}$ in Cu Powder

### 2.3.1. S vacancy Formation and $Cu^{2+}$ Dissolution Mechanism

Since $MoS_2$ is only soluble in aqua regia and heated concentrated sulfuric acid, it is insoluble in water, so the total concentration of Mo ion measured by AAS after 10 h of sonication of $MoS_2$ in aqueous solution is about 7.41 mg/L, and the leaching rate is about 2.47%; that is, the dissolution of the $MoS_2$ catalyst itself is negligible. In the process of copper dissolution, ultrasonic vibration generates energy to induce $MoS_2$ deformation and

then generates electrons and holes. These electrons contact and react with $O_2$ molecules in the water to generate $\cdot O_2^-$, and the holes react with $OH^-$ in the water to generate $\cdot OH$. There are vacant orbitals in the outer layer of the S atom, which can accept the external electron cloud, so the aqueous solution of the $MoS_2$ is acidic [21].

In this weakly acidic system, Cu elemental material is transformed into $Cu^{2+}$ by interaction with OH and $O_2^-$, which are two kinds of free radicals. In our experiment, the reaction solution was allowed to stand for 12 h, the supernatant was filtered with 0.22 $\mu$m filter head, and the filtrate obtained was pure solution of $Cu^{2+}$. The filtrate was evaporated at 60 °C to obtain new copper. The specific process is shown in Equations (1)–(5) below. Meanwhile, in order to verify that $MoS_2$ can generate piezoelectric potential under ultrasound drive, we chose two models with $100 \times 100 \times 5$ nm and forces of $10^4$ Pa (Figure 8a) and $10^6$ Pa (Figure 8b), respectively. We found that the greater the applied pressure, the greater the piezoelectric potential generated on the material surface; this indicates that deformation easily occurs with $MoS_2$ and that the applied pressure is proportional to the generated piezoelectric potential.

$$MoS_2 \text{ (polarize)} \rightarrow e^- + H^+ \tag{1}$$

$$O_2 + e^- \rightarrow \cdot O_2^- \tag{2}$$

$$H_2O + e^- \rightarrow \cdot H + OH^- \tag{3}$$

$$OH^- + H^+ \rightarrow \cdot OH \tag{4}$$

$$Cu + \cdot OH/O_2^- + H^+ \rightarrow Cu^{2+} + H_2O \tag{5}$$

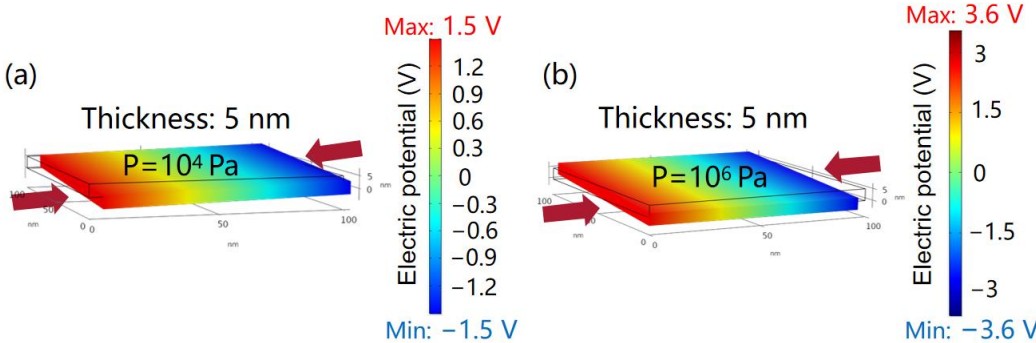

**Figure 8.** (**a**,**b**) COMSOL simulation analysis of catalysts.

An XPS test was used to analyze the changes in the valence states of each element in the catalyst before and after the reaction (Figure 9). As shown in Figure 9a, there was no peak of copper element in the full spectrum before the reaction. However, after the reaction, the diffraction peak of $Cu^{2+}$ with larger peak strength increased in the system (Figure 9b), which may be the result of the interaction of Cu with $\cdot OH$ and $\cdot O_2^-$ in the weakly acidic environment. After the reaction, there were unreacted elemental copper and $Cu^+$ generated due to insufficient oxidation capacity (Figure 9c). For the element carbon (Figure 9d), the three peaks were located at 287.1 eV, 285.9 eV, and 284.6 eV, corresponding to O=C–O, C=O and C=C/C–C, respectively [22]. In the high-resolution spectra of Mo 3d (Figure 9e), the catalyst shows a double peak at 232.9 eV and 229.7 eV, corresponding to Mo $3d_{3/2}$ and Mo $3d_{5/2}$ of $Mo^{4+}$ [23,24]. However, the binding energy at 226.7 eV is the result of S 2s of $S^{2-}$ in $MoS_2$ catalyst. The binding energies at 235.4 eV and 232.1 eV and 233.3 eV and 229.3 eV correspond to $Mo^{6+}$ and $Mo^{5+}$, respectively. The formation of $Mo^{6+}$ may be due to the oxidation of the catalyst, and then $Mo^{6+}$ is reduced to $Mo^{5+}$, but the Mo in the sample mainly exists in the form of $Mo^{4+}$. In the S 2p spectrum (Figure 9f), the double peaks located at 162.2 eV and 162.7 eV are caused by S $2p_{3/2}$ and S $2p_{1/2}$ in $S^{2-}$. The ligand or terminal $S^{2-}$ species of $S_2^{2-}$ bridge form a double peak at 163.5 eV and 163.8 eV [19], and the binding energy at 168.8 eV represents $SO_4^{2-}$.

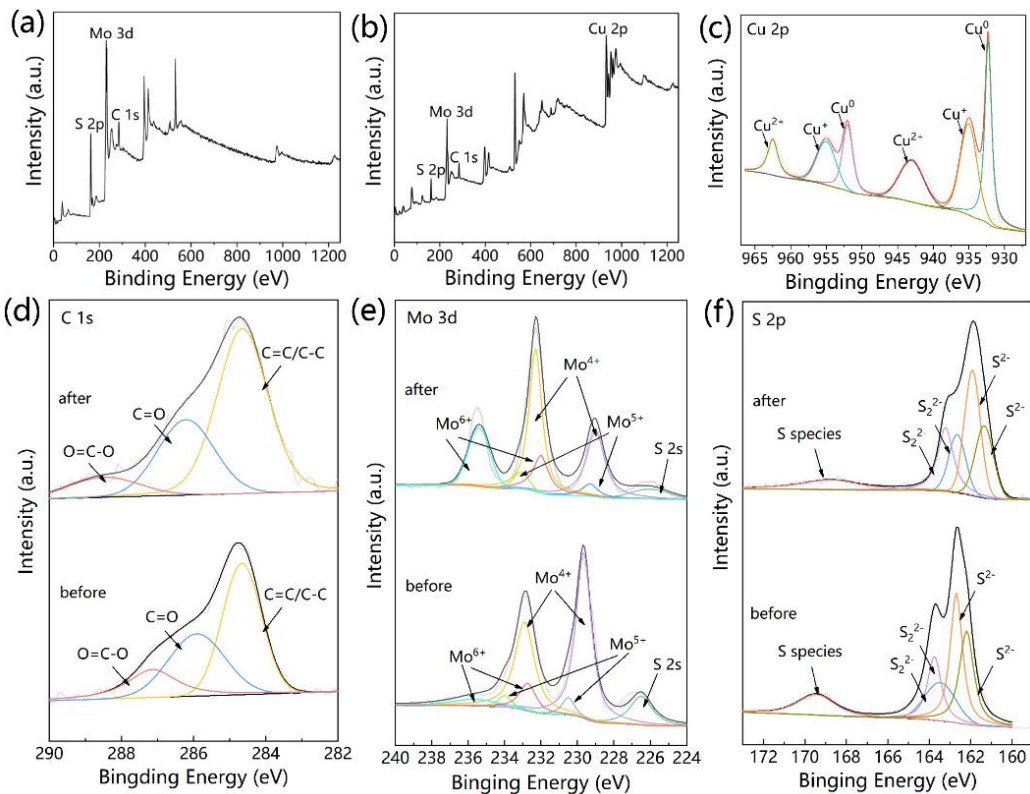

**Figure 9.** Changes in the binding energy of each element in the catalyst before and after the reaction. (**a**) full spectrum before reaction. (**b**) full spectrum after reaction. (**c**) fine spectrum of Cu 2p after reaction. (**d**) fine spectrum of C 1s after reaction. (**e**) fine spectrum of Mo 3d after reaction. (**f**) fine spectrum of S 2p after the reaction.

### 2.3.2. Free Radical Trapping Experiment

In this experiment, tert-butanol (TBA) and p-benzoquinone (BQ) were used as ·OH and ·$O_2^-$ to analyze the active species in the reaction system, as shown in Figure 10a. The results show that ·OH and ·$O_2^-$ promote the dissolution of $Cu^{2+}$ in the reaction process. The free radical trapping experiments show that a piezoelectric effect is generated by the $MoS_2$ catalyst under the action of ultrasonic vibration. The $e^-$ generated by piezoelectric action reacts with $O_2$ to form ·$O_2^-$; the $OH^-$ dissociated from $H_2O$ reacts with $H^+$ to form ·OH; and the elemental copper forms $Cu^{2+}$ under the action of ·OH and ·$O_2^-$ [25,26]. In order to verify the type of active oxide produced in the reaction system, we carried out an EPR test. According to Figure 10b,c, there are ·$O_2^-$ and ·OH free radicals in the reaction system, and the concentration of the two free radicals in the system increases significantly with the increase in reaction time, which well verifies the free radical capture experiment.

### 2.3.3. Dissolution Effect of $MoS_2$ Catalyst on Copper in Circuit Board

The SEM of the sample after ultrasonic action on the circuit board is shown in Figure 11. The experimental conditions changed are the ultrasonic reaction time or the addition of a catalyst or not. Figure 11a,b show the low power (50×) and high power (5000×) field emission scanning electron microscopy photos of the original circuit board, respectively. It can be seen from Figure 11b that there are slight cracks on the surface of the circuit board, which may be caused by the manufacturing or transportation of the circuit board. In Figure 11c,d, $MoS_2$ was used as the piezoelectric catalyst, and the ultrasonic action was performed at 300 W power for 1 h. The crack was gradually obvious. Different from Figure 11b,d, the width and depth of the crack increased significantly. Figure 11e,f shows the circuit board of ultrasonic reaction for 5 h. It can be seen intuitively that the cracks have been turned into holes, indicating that a considerable amount of Cu has been oxidized

to $Cu^{2+}$ by $\cdot OH$ and $\cdot O_2^-$. As for Figure 11g,h, the number of holes and holes increased significantly and became dense, indicating that more elemental Cu was involved in the piezo-catalyzed reaction than in Figure 11e,f. Figure 11i,j were treated with ultrasound for 10 h without catalyst. Compared with the high-power scanning electron microscopy photos, it can be seen that the effect was between 1 h and 5 h under the condition of adding catalyst. Figure 11k,l were stirred for 10 h with catalyst added, and the speed of the magnetic stirrer was 400 r/min. It can be seen that its effect was weaker than that of ultrasonic catalyst added for 1 h. Comparing the experimental conditions in Figure 11, it can be seen that under the above reaction conditions the reaction effect is best when the catalyst is added for 10 h under ultrasonic action. As can be seen from Figure 11m, with the increase in ultrasonic time, the color of the circuit board surface gradually deepens, which means that the metal copper attached to the circuit board is gradually corroded, and more and more $Cu^{2+}$ are dissolved in the solution; this is entirely consistent with the test results of SEM. At the same time, we used the AAS method to determine that the concentration of the $Cu^{2+}$ dissolved from the circuit board after 10 h ultrasonic action of synthetic $MoS_2$ catalyst is 53.33 mg/L.

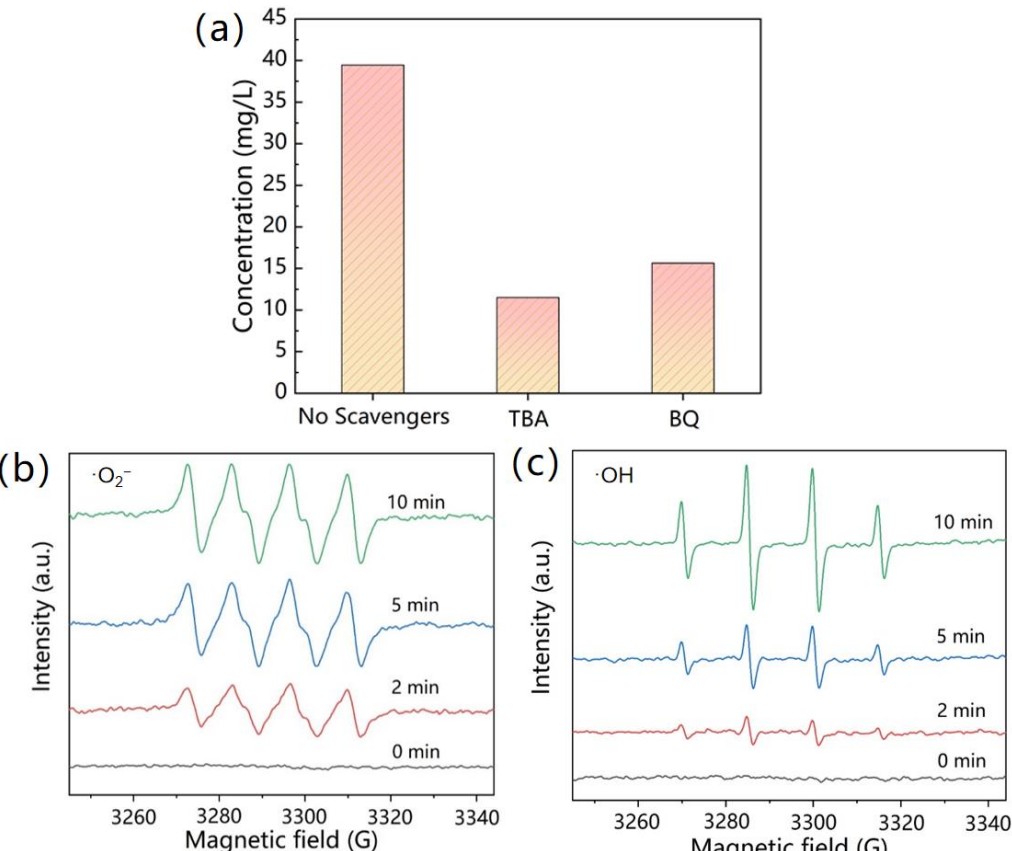

**Figure 10.** (**a**) free radical trapping experiment: TBA → $\cdot OH$; BQ → $\cdot O_2^-$. (**b**) Spectra of superoxide radical in EPR test. (**c**) Spectra of hydroxyl in EPR test.

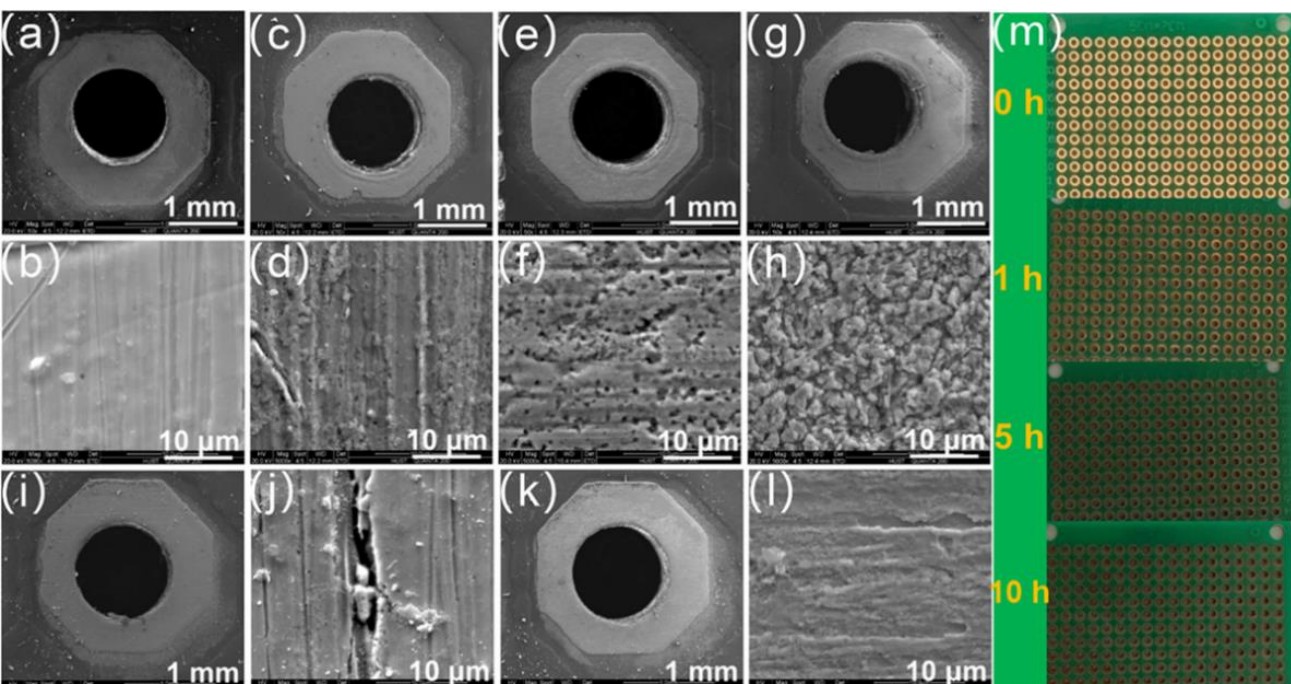

**Figure 11.** SEM images of circuit boards under different ultrasonic times: (**a**,**b**) sonicate for 0 h with catalyst. (**c**,**d**) sonicate for 1 h with catalyst. (**e**,**f**) sonicate for 5 h with catalyst. (**g**,**h**) sonicate for 10 h with catalyst. (**i**,**j**) sonicate for 10 h without catalyst. (**k**,**l**) stir for 10 h with catalyst. (**m**) physical map of the circuit board.

## 3. Materials and Methods

### 3.1. Preparation of $MoS_2$ by Hydrothermal Method

$MoS_2$ was synthesized by a simple hydrothermal method as follows: 0.72 g $Na_2MoO_4 \cdot 2H_2O$ and 0.69 g $CH_4N_2S$ powder were dissolved in 1 mL of 1M $C_8H_{15}ClN_2$, denoted as A. One mL HCl was diluted in 60 mL deionized water, and the dilute solution was slowly dropped into A and stirred continuously for 12 h until it became a clear and transparent solution, denoted as B. B was transferred to a 100 mL autoclave and kept at 220 °C for 24 h. The reactor was cooled to room temperature, and the synthesized products were rinsed in distilled water 3~5 times. After drying at 60 °C in an oven, black $MoS_2$ powder was obtained.

### 3.2. Characterization and Analysis Methods

In this study, XRD (D8 Advance XRD) was used and Kα rays of Cu target were taken as the radiation source. The working voltage was 40 kV, the current was 50 mA, the scanning range was 5–75°, and the step length was 0.02 °/min. The material composition, crystal type, crystal plane data, and lattice fringes of the samples were characterized. Using SEM (Hitachi, Tokyo, Japan, S-4800 scanning electron microscope), scanning acceleration voltage 200 kV, magnification of 50 to 1.1 million times, and TEM (FEI Company, Hillsboro, OR, USA, FEI Tecnai G2 F30 transmission electron microscope), scanning acceleration voltage 200 kV operating conditions, the morphology and structure of the sample surface were obtained. AFM (SPM 9700) was used to obtain the thickness information of the samples. A BET test (MAC, ASAP 2020) was used to obtain the specific surface area, pore volume, pore size distribution, and nitrogen adsorption and desorption curve data of the samples under the working conditions of 2.0 nm resolution, 120 kV acceleration voltage, and 0.2–6 μm beam spot size. XPS technology (Thermo Scientific, Waltham, MA, USA, Escalab 250Xi) was used to analyze all the elements except He with Kα of Mg target as X-ray source. The elemental content, elemental chemical valence, and orbital electron binding energy of the sample surface were analyzed at an energy resolution of 0.45 eV (the binding energy of C 1s at 284.8 eV was used as a standard to correct the binding energy of the other elements).

Confocal microscopic Raman spectroscopy (Mack Corporation, Arlington, VT, USA, ASAP 2460) was used to obtain the molecular structures in the samples.

In this study, COMSOL software was used to simulate the distribution of the voltage potential. When a sound pressure of $10^6$ Pa is applied perpendicular to the plane, an obvious electric field is generated and distributed on the upper and lower sides of the model. When a sound pressure of $10^6$ Pa is applied parallel to the plane, an obvious electric field is generated and distributed on the side of the model.

*3.3. Piezoelectric Catalysis in the Dissolution of Copper from Circuit Boards*

0.05 g $MoS_2$ was weighed in a 250 mL beaker, and 100 mL deionized water was added. Three circuit boards with a mass of about 4.74g were placed in it and reacted in the ultrasonic pool with a power of 300 W. After 1 h, 5 h, and 10 h, they were taken out to be tested.

The concentration of Cu ion was detected by the AAS method: 0.1 g Cu powder and 0.05 g $MoS_2$ were weighed into a 250 mL beaker, then 100 mL deionized water was added and treated with 300 W ultrasound for 10 h. Then 5 mL was filtered by a 0.22 μm filter membrane into a centrifuge tube, and the concentration of Cu ion was detected by the AAS method.

Our determination of the concentration of the Cu ion: In this study, in addition to using the AAS detection method to determine the concentration of the Cu ion, we also used 2, 9-dimethyl-1, 10-phenanthroline spectrophotometry. The specific methods were as follows: copper standard concentration solutions of 0, 25, 50, 100, 150, and 250 μg/mL were prepared, and the corresponding absorbance values were measured by 2, 9-dimethyl-1, 10-phenanthroline spectrophotometry. The correlation coefficient $R^2$ of the fitted curve was 0.9999.

After the reaction solution was filtered through 0.22 μm filter membrane, 15 mL was taken accurately in a 25 mL colorimetric tube. Then 1.5 mL hydroxylamine hydrochloride solution, 3 mL sodium citrate solution, 3 mL sodium acetate solution, and 1.5 mL neocuproine solution were added in turn and shaken well. Next, water was added to the mark, the solution was shaken well, and the absorbance of Cu ion was measured at 457 nm with a spectrophotometer after standing for 5 min. The absorbance of the Cu ion was measured by spectrophotometer at 457 nm and recorded as A. After shaking well, water was added to the marked line and the solution was shaken well again and let stand for 5 min. Then an A spectrophotometer was used to measure the absorbance of the Cu ion at 457 nm wavelength, denoted as A. Finally, the concentration of the Cu ion was calculated according to Equation (6):

$$C = \frac{A - A_0 - a}{bV} \tag{6}$$

where A is the measured absorbance value of $Cu^{2+}$, $A_0$ is the blank background value, a is the intercept of the regression equation on the Y-axis, b is the slope, and V is the volume of samples added during the test.

## 4. Conclusions and Prospect

In this study, $MoS_2$ synthesized by the hydrothermal method was used as the piezoelectric catalytic material, and the effect of piezoelectric catalysis on the leaching of metallic copper from copper powder was investigated. The $\cdot OH$ and $\cdot O_2^-$ generated by the process promoted the leaching of copper ions, the concentration of the dissolved $Cu^{2+}$ was 39.46 mg/L after 10 h of reaction, and the leaching rate of the Mo ions was only 2.47%. A small pilot experiment was also simulated with the circuit board as material. During the ultrasonic vibration, the vibration stripped off a part of Cu that was tightly attached to the circuit board so that the stripped Cu was dispersed in the $MoS_2$ solution. In addition, the active oxide produced by the process could oxidize Cu monomers to $Cu^{2+}$, and the concentration of $Cu^{2+}$ in the solution was detected by AAS as 39.46 mg/L. The SEM images clearly show the morphology of the circuit board after 0 h, 1 h, 5 h, and 10 h. The Cu

attached to the circuit board was gradually oxidized by ·OH and ·$O_2^-$ radicals, and the concentration of $Cu^{2+}$ in the solution was 53.33 mg/L after 10 h. This study combines piezoelectric catalysis with metal dissolution from e-waste for the first time on the basis of environmentally friendly catalytic materials. There is a degree of metal dissolution from the e-waste as well as a degree of copper dissolution, which further elucidates the mechanism of precious metal dissolution using piezoelectric catalysis and also provides new ideas for the recovery of precious metals in e-waste using piezoelectric catalysis technology. Compared with research on the recovery of precious metals by chemical agents such as strong acid, alkali, and fluoride, the metal dissolution effect in this study needs to be further improved. In the future, the metal dissolution efficiency can be improved by optimizing the experimental conditions. In addition, the ultrasonic vibration causes the catalyst to be dispersed in the solution very evenly, which makes recovery of the catalyst difficult. In addition, the circuit board used in this study contains copper only, and the dissolution and recovery of Au, Ag, Pt, and other metals in the circuit board and other precious metals in the ore will be a hot topic for further research to meet the needs of practical application.

**Author Contributions:** Conceptualization, H.L.; methodology, H.L. and F.C.; validation, H.L. and F.C., investigation, validation, supervision, writing—review and editing, H.Y.; writing—original draft preparation, H.L.; writing—review and editing, F.C. and X.H. All authors have read and agreed to the published version of the manuscript.

**Funding:** This research received no external funding.

**Data Availability Statement:** The data presented in this study are available on request from the corresponding authors.

**Conflicts of Interest:** The authors declare no conflict of interest.

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
