# Peer review of "Performance and Mechanism of Hydrothermally Synthesized MoS2 on Copper Dissolution"

_catalysts, doi:10.3390/catal13010147_

Round 1

Reviewer 1 Report

Lu reported MoS2 for the dissolution of metal copper. The ultrasound power triggered free radicals, which participated the oxidation of Cu0 to Cu2+. This work extends the application of piezocatalysis. Major revisions are suggested as follows:

1.     There are a lot of format errors, for example:

(a)Line 43, 46, format error. The and Space.

(b)Line 50 is confusing. Line 72, space. Line 115, space of 0.05g. Line 127 R2

(c)Line 74, define low costs.

2.     Delete figure 1 and figure 2 or remove it to supporting information. Figure 2 caption, added ppm or mg/L after concentration

3.     Line 158, format error. Line 161, revise the figure caption

4.     Line 164, Figure a-b should be Figure 4a-b.

5.     Line 186, reorganize Figure 5. Give clear explanation of Figure 5b and 5c.

6.     Figure 6, error in BET surface error.

7.     XPS part should be reorganized. When it first appears, do not mention any contents of Cu. This part belongs to the characterization of MoS2.

8.     Line 224, “The effect of catalyst on Cu2+” is confusing. The author wants to dissolve Cu, but the content described at the beginning of this part should be reorganized.

9.     Added the discussion of Figure 8a.

10.  Line 268, “Influence of catalyst on Cu2+ dissolution” is hard to understand. Explain Figure 10.

11.  Line 276, define the S vacancy. How to prove it?

12.  Put Figure 12 after Figure 13.

13.  The conclusion part should be rewritten.

Reviewer 2 Report

The authors reported a hydrothermal method to synthesize MoS2 for the application of recycling copper. I didn't see much of novelty of the article. As stated in the introduction, the recycling of noble metal in the circuit board is more important.

Second, the so-called piezo-electric catalysis process stated in the article is very vague to me. The mechanism is vague. Either sonicating or simply heating up the Cu powder in acidic solution could convert Cu into Cu2+. I think the topic is out of the scope of the journal.

Last but not least, the manuscript is full of grammar mistakes, and the lack of citation. The author compare their hydrothermally-synthesized MoS2 with the so-called commercial MoS2 and showed the poor morphology and performance of "commercial MoS2" but I didn't find the source of this commercial product in the article.

Therefore, I would recommend that the editor reject the paper without further consideration. I think the authors should consider submitting the article to the journal that fits the topic of characterizing the properties of MoS2.

Reviewer 3 Report

The authors presented for the first time the combination of the piezoelectric catalysis technology with the recovery of precious metals from electronic waste, with a certain effect of precious metal dissolution, further elucidating the use of piezoelectric catalysis for precious metal dissolution mechanism, and providing a new idea for the recovery of precious metals from electronic waste. The presented paper is interesting and I do recommend to be published after major revision due to the comments that follows:

Comment 1:

Some parts of the text are written in past and some in the present tense. Please perform a detailed check through the whole document of the English grammar, spacing, units, etc.

Comment 2:

The „Preparation of MoS2 by hydrothermal method” section should be re-written for better understanding what is denoted as “A” and what is “B”.

Comment 3:

The Figure 1 was not mentioned in the text

Comment 4:

Please check and correct the language, line numbers 134-141.

Comment 5:

Line numbers 142 and 143. The “A” is denoted as the measured absorbance value of Cu2+, and the intercept of the regression equation on the Y-axis. The B should be written as “b”. Please correct.

Comment 6:

Line 165. The figure number is missing.

Comment 7:

Figure 5b and c are not mentioned in the text.

Comment 8:

The whole document needs to be checked for English grammar, while the characterization methods are not adequately and fully described.

Round 2

Reviewer 1 Report

The issues have been addressed. This work is publishable under current state.